# Elongator and codon bias regulate protein levels in mammalian peripheral neurons

Joy Goffena[1], Frances Lefcort[2], Yongqing Zhang[3], Elin Lehrmann [3], Marta Chaverra[2], Jehremy Felig[1], Joseph Walters[1], Richard Buksch[1], Kevin G. Becker[3] & Lynn George[1]

Familial dysautonomia (FD) results from mutation in *IKBKAP/ELP1*, a gene encoding the scaffolding protein for the Elongator complex. This highly conserved complex is required for the translation of codon-biased genes in lower organisms. Here we investigate whether Elongator serves a similar function in mammalian peripheral neurons, the population devastated in FD. Using codon-biased eGFP sensors, and multiplexing of codon usage with transcriptome and proteome analyses of over 6,000 genes, we identify two categories of genes, as well as specific gene identities that depend on Elongator for normal expression. Moreover, we show that multiple genes in the DNA damage repair pathway are codon-biased, and that with Elongator loss, their misregulation is correlated with elevated levels of DNA damage. These findings link Elongator's function in the translation of codon-biased genes with both the developmental and neurodegenerative phenotypes of FD, and also clarify the increased risk of cancer associated with the disease.

[1] Department of Biological and Physical Sciences, Montana State University Billings, Billings, MT 59101, USA. [2] Department of Cell Biology and Neuroscience, Montana State University, Bozeman, MT 59717, USA. [3] Gene Expression and Genomics Unit, National Institute on Aging, National Institutes of Health, Baltimore, MD 21224, USA. Correspondence and requests for materials should be addressed to L.G. (email: lynn.george@msubillings.edu)

The genetic code consists of trinucleotide units termed codons, with each of these units encoding a specific amino acid. In all species of life, the same amino acid can be encoded by up to six different trinucleotide combinations, or synonymous codons. The reason for this redundancy is not well understood, although, interestingly, synonymous substitutions can impair protein function[1]. Codon usage has recently been spotlighted as a key determinant of translation elongation rates and co-translational protein folding, with preferred codons enhancing translational efficiency and folding fidelity[2,3]. The unequal usage of synonymous codons is referred to as codon bias and the universal nature of this bias, from yeast to humans, suggests the existence of a secondary code within the more familiar genetic code. This secondary code is emerging as a major regulator of translational speed and co-translational protein folding and thereby a significant determinant of the cellular levels of specific proteins.

A key component of the cellular machinery used to interpret codon bias is the molecular complex Elongator, composed of 6 subunits (ELP1–6), which chemically modifies transfer RNAs (tRNAs). Within the ribosomal A site, the tRNA anticodon loop forms Watson–Crick base pairs with its cognate messenger RNA codon. Flexibility at the first anticodon position allows for non-standard (wobble) pairing and, in particular, uridines in the wobble position ($U_{34}$) are capable of recognizing multiple nucleotides. Specifically, lysine, glutamine, and glutamic acid are each encoded by two trinucleotide combinations that end in either AA or AG (AAA and AAG = Lys; CAA and CAG = Gln; GAA and GAG = Glu), and a single eukaryotic tRNA with a U in position 34 can recognize both the AA- and AG-ending (wobble) versions of these codons (Fig. 1). Chemical modifications to $U_{34}$, including the addition of both a thiol ($s^2$) and a methoxy-carbonyl-methyl ($mcm^5$) refine $U_{34}$ recognition and specifically enhance the translational efficiency of problematic AA-ending codons that impose restrictive codon–anticodon interactions (Fig. 1)[4–8]. The impact of the $mcm^5s^2$ modification on the translation of the AG-ending (wobble) codons for these amino acids is not completely understood, with some studies suggesting it has no impact and others indicating that it decreases AG translational efficiency[4,9–12]. Multiple studies in a variety of model organisms, as well as in brain tissue and fibroblasts from patients with familial dysautonomia (FD) that carry mutations in IKBKAP/ELP1, have shown that Elongator is essential for the synthesis of the $mcm^5$ modification[13–18]. In addition, thiolation of the wobble uridine depends on the addition of the same $mcm^5$ moiety, such that neither modification is present in the absence of Elongator[19]. Importantly, in Saccharomyces cerevisiae and Caenorhabditis elegans, loss of tRNA $U_{34}$ modification leads to widespread protein aggregation and imbalanced protein homeostasis[3], and multiple studies in fission yeast have demonstrated that proteins encoded by AA-biased transcripts are depleted in Elongator (Elp) mutants[19–21].

To gain a better understanding of the role that codon bias may have in mammalian neurons, we conducted a comprehensive analysis of protein levels in the dorsal root ganglia (DRG) of mice in which Ikbkap/Elp1 is conditionally ablated in the peripheral nervous system (PNS)[22]. Although loss of any of the Elongator subunits is sufficient to disrupt function of the complex[13], we used an Ikbkap/Elp1 knockout, because mutations in the human gene cause FD[23]. The primary IKBKAP mutation in FD causes tissue-specific exon skipping, mRNA nonsense-mediated decay, and reduced levels of the encoded IKAP/ELP1 protein, with the PNS being the most gravely impacted[23–26]. In Wnt1-Cre; Ikbkap$^{LoxP/LoxP}$ conditional knockout (CKO) mice, Ikbkap expression is selectively ablated in the PNS[22]. These CKO mice exhibit reduced levels of $mcm^5s^2U$ tRNA modification (Supplementary Fig. 1) and recapitulate many FD hallmarks including a depleted number of pain and temperature-sensing TrkA+ neurons in the DRG, a primary site of Ikbkap expression[22,25]. By comparing protein levels within the DRG of CKO and control mice, we demonstrate that the differential usage of AA- and AG-ending codons, in coordination with the Elongator complex, function together to regulate the levels of particular proteins in the mammalian PNS that are ultimately essential for neuron survival.

## Results

**Codon-biased eGFP is misexpressed in CKO neurons.** To directly investigate a role for the Elongator complex in the translation of codon-biased transcripts in mammals, we engineered a codon-biased version of enhanced green fluorescent protein (eGFP) that exclusively uses AA-ending codons for the amino acids Lys, Gln, and Glu, and compared the expression level of this construct with mammalian optimized eGFP in which 42 out of 44 codons for these same amino acids end in AG. In neuron cultures from both control and CKO embryos, the AA-biased eGFP was expressed at a significantly lower level than the AG-biased construct (Fig. 2), indicating that even in the presence of the $mcm^5s^2$ modification, restrictive AA codon–anticodon interactions impede protein production[4–8]. Somewhat surprisingly, there was no difference in the amount of this reduced expression between CKO and control neurons. This may have been due to the small size of the eGFP gene (see below). Unexpectedly however, AG-biased eGFP was expressed at a significantly higher level in CKO neurons compared with controls (Fig. 2). This raises the possibility that the $mcm^5s^2$ modification present in the control may indeed reduce the ability of $U_{34}$ to pair with G-ending (wobble) codons, such that the absence of the modification actually increases the efficiency of AG translation[9–11], although further kinetic data are required to show this.

**Codon bias and mRNA size influence Elongator dependence.** Across both the mouse and human genomes, AG-ending codons for Lys, Gln, and Glu are preferentially used over AA-ending codons (~64% of all such codons end in AG). This bias is often measured as the number of AA-ending codons over the number

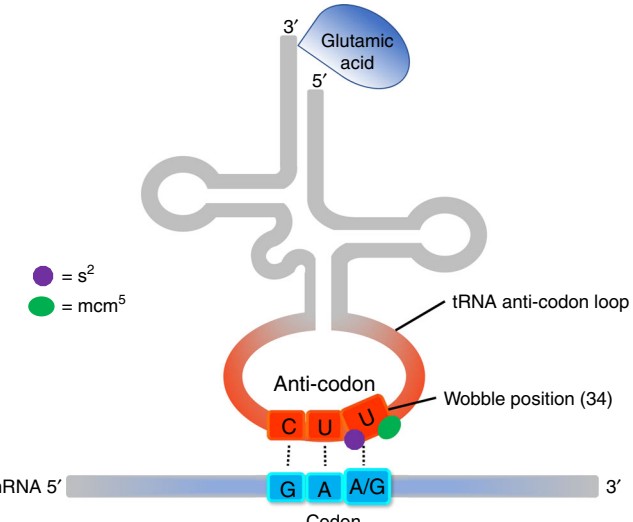

**Fig. 1** The wobble uridine ($U_{34}$) of tRNA molecules that recognize both AA- and AG-ending codons for Lys, Gln, and Glu, is modified by the addition of both a thiol ($s^2$) and a methoxy-carbonyl-methyl ($mcm^5$). This double modification enhances the translational efficiency of AA-ending codons

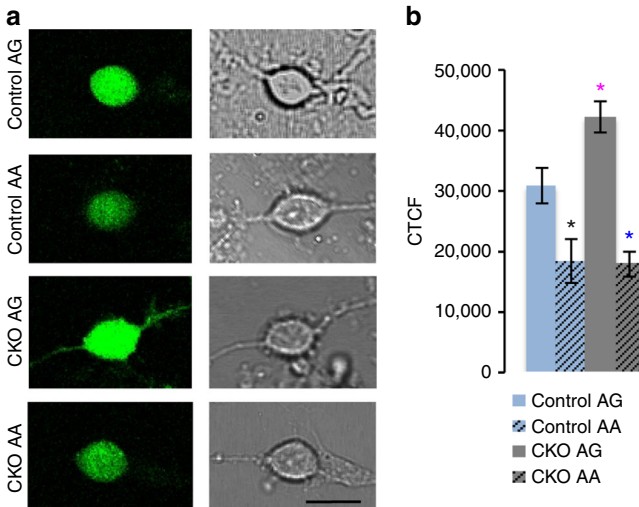

**Fig. 2** Expression of codon-biased reporter constructs in CKO neurons. **a** Fluorescent and transmitted images of control and CKO neurons expressing AA- and AG-biased eGFP. **b** Quantification of corrected total cell fluorescence (CTCF); *P*-values: AG-biased versus AA-biased: black asterisk, *control = 0.011; *blue asterisk, *CKO = < 0.001. AA-biased control versus AA-biased CKO = 0.89; *pink asterisk, *AG-biased control versus AG-biased CKO = 0.008; error bars denote SEM; scale bar: 6 μm

of AG-ending codons, or the AA:AG ratio, with a ratio of 1 indicating an unbiased usage. In mouse and humans, the average AA:AG ratio across all open reading frames is 0.67 and 0.76, respectively (ftp://ftp.ncbi.nlm.nih.gov/pub/CCDS/). To visualize this bias toward the use of AG-ending codons, we plotted the total number of coding sequences as a function of AA:AG ratio. A total of 84.1% of all coding sequences in mouse and 74.2% in humans exhibit an AA:AG ratio < 1 (Fig. 3a). To further investigate a mammalian role for Elongator in the translation of codon-biased genes, we conducted a comprehensive proteomics analysis. DRG from control and CKO embryos were pooled, respectively (*n* = 7 embryos per genotype), and the proteins subjected to ultra-high-pressure liquid chromatography tandem mass spectrometry (UHPLC–MS/MS). To identify genes that were specifically misregulated at a posttranscriptional level, we overlaid the resulting proteome data with a transcriptome analysis of genes expressed in these same tissues (E17.5 DRG) (GSE80130) and filtered the combined data sets to identify genes that were detected in both the transcriptome and proteome analyses, and that were transcribed normally (average fold change values < 1.5 and > − 1.5). Genes that were detected in one sample type (either control or CKO), and not detected in the counterpart sample, were omitted. The remaining 6,448 genes (Supplementary Data 1) were then analyzed for codon usage. Total transcript length in number of codons was also considered.

Using this data set, we analyzed the behavior of protein expression levels as a function of codon bias and found that increasing AA bias correlates with an increasing likelihood that a protein will exhibit decreased expression (fold change ≤ − 2) (Fig. 3b and Supplementary Data 2). Importantly, we also found that large AA-biased genes are more likely to be impacted by Elongator loss than are small AA-biased genes, presumably because large genes contain the highest numbers of AA-ending codons. In mouse and humans, the average mRNA coding sequence consists of 556 and 572 codons, respectively (https://www.ncbi.nlm.nih.gov/CCDS/CcdsBrowse.cgi). As shown in Fig. 3b, 55.0% of proteins encoded by transcripts ≥ 1,755 codons (5,265 bp) with an AA:AG ratio > 1.5 are depleted in the CKO, as compared with only 26.5% of proteins with the same AA:AG

ratio, but a transcript length ≥ 505 codons. Considering the small size of the eGFP coding sequence (240 codons), these data may explain why the expression of AA-biased eGFP was comparable between control and CKO neurons in our in vitro experiments (Fig. 2).

Proteins expressed normally (fold change > − 2 and < 2) were also analyzed as a function of increasing AA-bias. Figure 3c shows that the ability to maintain normal expression levels becomes increasingly dependent on Elongator as AA-bias and transcript length increase. To assess the impact of total AA number in the context of no bias, we also plotted the percentage of depleted proteins as a function of increasing AA number, irrespective of the AA:AG ratio. When transcripts of all AA:AG ratios are considered, the percentage of depleted proteins is consistently about 27%, regardless of AA number (Fig. 3d). These data indicate that AA-bias (a high number of AA-ending codons in the context of a low number of AG-ending codons), rather than AA number, determines Elongator dependence.

We next examined the impact of Elongator loss on the expression levels of AG-biased genes. Interestingly, we found that the percentage of proteins that are upregulated (fold change ≥ 2) increases with increasing AG-bias (Fig. 3e) (Supplementary Data 2). These data suggest that the presence of mcm⁵s² in control neurons decreases the translational output of AG-ending codons, whereas in Elongator knockout cells that lack the modification, highly AG-biased transcripts are translated more efficiently. These data also show that small, AG-biased transcripts are more likely to be impacted by Elongator loss than are large, AG-biased transcripts. This conclusion supports our in vitro observation that the expression of AG-biased eGFP, at only 240 codons, increases in the absence of Elongator (Fig. 2). These data are also consistent with a study in yeast *elp3* mutants showing small decreases in ribosome occupancy for AG-ending codons[27] and support the hypothesis that an mcm⁵s² modification may reduce the ability of $U_{34}$ to pair with G-ending codons[9–11]. In addition, these data also explain why AA abundance in the context of an equally high number of AG-ending codons did not have an impact on protein levels in our study (Fig. 3d); the decreased translation rate of AA-ending codons may have been offset by an increased translation rate of an equally large number of AG-ending codons.

In addition to the nervous system, *Elp* genes are expressed somewhat ubiquitously in numerous other tissue types[24]. To gain insight into the frequency with which codon bias might be generally used in the regulation of protein levels within mice and humans, we also filtered both ORFeomes with parameters that are indicative of Elongator dependence (Fig. 3). Approximately 3.5% of mouse coding sequences, and 5% of human coding sequences fall within our most stringent parameters (a transcript size ≥ 1,755 total codons with an AA:AG ratio > 1.3, or a transcript size ≤ 150 total codons with an AA:AG ratio < 0.3) (Supplementary Data 3). Approximately 12% of mouse and 18% of human coding sequences fall within slightly less stringent size parameters (a transcript size ≥ 1,005 total codons with an AA:AG ratio > 1.3, or a transcript size ≤ 300 total codons with an AA:AG ratio < 0.3) (Supplementary Data 4). These data suggest that codon bias serves as a basic, somewhat pervasive mechanism for regulating protein levels within mammals.

**Large, AA-biased Elongator targets**. Having demonstrated that transcript length and AA:AG ratio regulate protein levels in an Elongator-dependent manner, we next investigated the identity of specific Elongator targets. For large, AA-biased genes, the proteome data set of normally transcribed genes was filtered for targets that consisted of ≥ 1,755 total codons with an AA:AG ratio

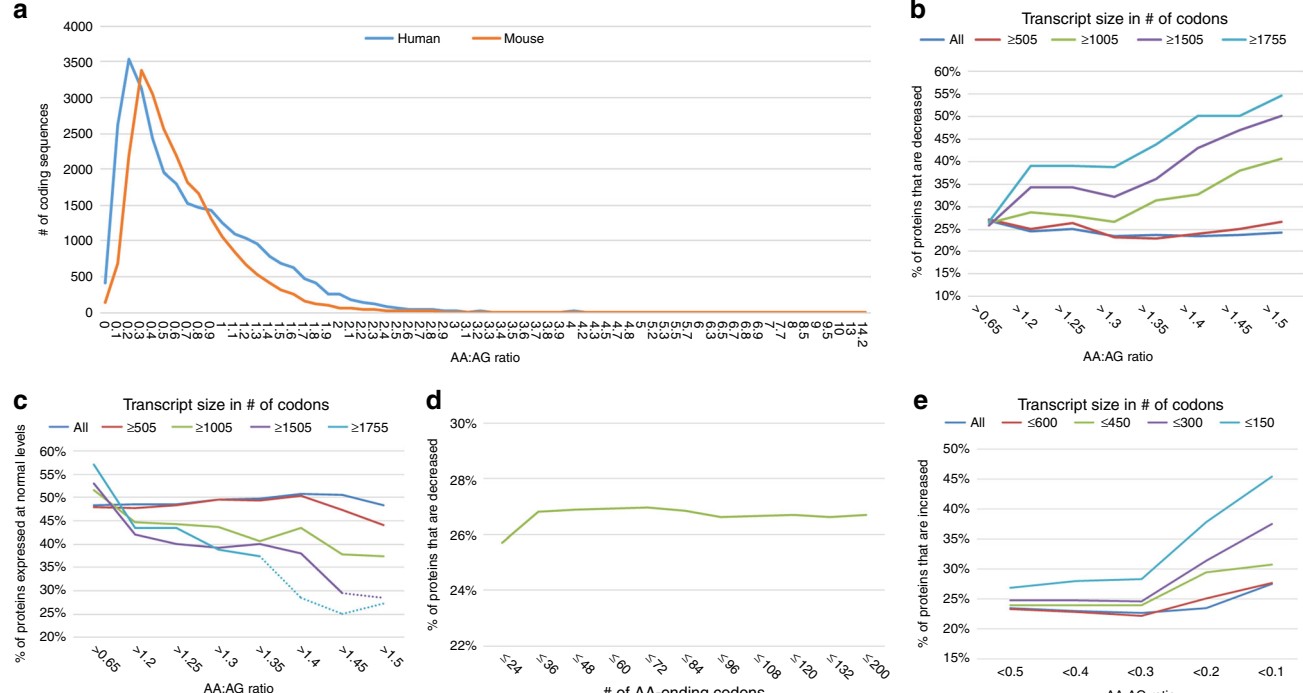

**Fig. 3** AA:AG ratio and transcript size dictate Elongator dependence. **a** Plotting the number of genes as a function of AA:AG ratio shows that 84.1% of mouse and 74.2% of human coding sequences preferentially use AG-ending codons (AA:AG ratio < 1) for Lys, Gln, and Glu. Each x axis value represents a minimum of one one gene. **b–e** Proteome comparison of normally transcribed DRG genes from E17.5 control and CKO embryos. Transcript size in number of codons is indicated by line color. **b**, **c** In the absence of Elongator, the percentage of proteins that are under-expressed (fold change ≤ − 2) increases with increasing AA bias and transcript length **b**, whereas the percentage expressed normally (fold change < 2 and > − 2) decreases **c**. **d** AA abundance in the absence of AA bias does not impact protein levels. **e** With increasing AG bias and decreasing transcript length, the percentage of proteins that are upregulated (fold change ≥ 2) increases. Dashed lines represent data calculated from fewer than five proteins. For numerical data and individual proteins, see Supplementary Data 2

> 1.3 (Fig. 3b). Of the 18 genes that remained, 7 were under-expressed (protein fold change ≤ − 2), 7 were expressed normally (protein fold change < 2.0 and > − 2.0), and 4 were overexpressed (protein fold change ≥ 2.0) (Supplementary Data 5). The reduced expression of less than half of these large, heavily AA-biased genes suggests that other posttranscriptional regulatory pathways (such as ubiquitination) may work to counteract the effects of Elongator loss for some genes. Alternatively, other as of yet unidentified gene characteristics may work in concert with codon bias and transcript size to mediate full Elongator dependence. Of the 7 under-expressed genes, *Brca2* and *Vcan* consist of over 3,300 total codons with 425 (*Brca2*) and 374 (*Vcan*) of those codons ending in AA (average number of AA-ending codons = 36 in mouse and 40 in humans) and an AA:AG ratio > 1.5, making them particularly compelling as potential candidates for Elongator dependence. Quantitative reverse transcriptase-PCR (RT-PCR) confirmed normal transcription of both genes in the CKO and quantitative immunohistochemistry (IHC) confirmed diminished levels of both proteins in DRG neurons (Fig. 4 and Supplementary Fig. 2).

*Brca2* has a key role in homology-directed repair (HDR) of DNA double-strand breaks[28]. Interestingly, our proteome analysis indicated the possible misregulation of multiple other DNA damage response genes, with some of those genes, including *Setx* and *Rbbp8*/*Ctip*, being strong candidates for direct Elongator regulation (Supplementary Data 6). As mutations in the human *SETX* gene cause multiple neurodegenerative disorders[29–32], we used quantitative IHC and RT-PCR to confirm decreased SETX in the context of normal transcription (Supplementary Fig. 3).

HDR is specifically required by neural progenitor cells during proliferation[33]. To investigate whether Elongator loss and the

resulting misregulation of *Brca2* and other DNA damage response genes leads to elevated levels of DNA damage during neurogenesis, we performed comet assays on control and CKO DRG at E12.5, during the peak of sensory progenitor proliferation[34,35]. As shown in Figure 4, *Wnt1- Cre; Ikbkap*^*LoxP/LoxP* embryos indeed exhibit significantly elevated levels of fragmented DNA at E12.5. Elevated DNA damage is also associated with amyotrophic lateral sclerosis (ALS)[36], a disease that strictly affects post-mitotic neurons that no longer have access to HDR. As mutations in the Elongator subunit *ELP3* are associated with ALS, we also analyzed the codon usage of genes required for non-homologous end-joining (NHEJ), the primary DNA repair pathway used by post-mitotic cells (Supplementary Data 7). Three genes, *Rif1*, *Pola1*, and *Prkdc*, fall out as likely targets for downregulation in the absence of Elongator. In particular, *Rif1* is a compelling candidate in that it is required for NHEJ and for suppression of the error-prone microhomology-mediated end joining pathway (MMEJ)[37].

**Small, AG-biased Elongator targets**. We also investigated the identity of small, AG-biased Elongator targets. By filtering normally transcribed genes for an AA:AG ratio < 0.2 and transcript length of ≤ 150 codons, we narrowed our target pool to 49 genes, with 19 being upregulated, 27 expressed normally, and 3 being downregulated (Supplementary Data 8). Of the 19 upregulated genes, 9 encode different H2A histones (both type 1 and type 2), with each comprising fewer than 150 total codons including 2 or fewer AA-ending and 23–29 AG-ending (AA:AG ratios between 0.00 and 0.08). *Hist1h2ab* and *Hist2h2aa1* were chosen as type 1 and type 2 representatives to verify normal transcription and

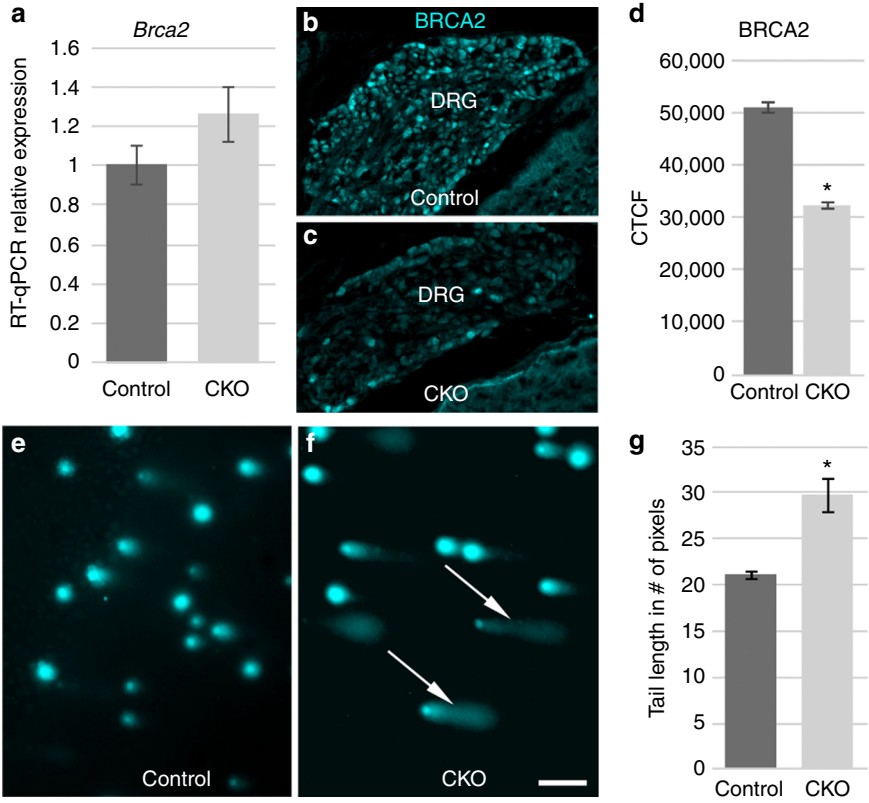

**Fig. 4** Depleted levels of BRCA2 are correlated with elevated levels of DNA damage. **a** RT-qPCR confirms normal levels of *Brca2* transcript ($P = 0.06$). **b–d** Quantitative immunohistochemistry confirms decreased levels of BRCA2 protein ($P < 0.001$). **e–g** Longer comet tails in the CKO show fragmented DNA ($P = 0.009$); error bars denote SD in **a** and SEM in **d**, **g**; scale bar: **b**, **c** 40 μm; **e**, **f** 12 μm

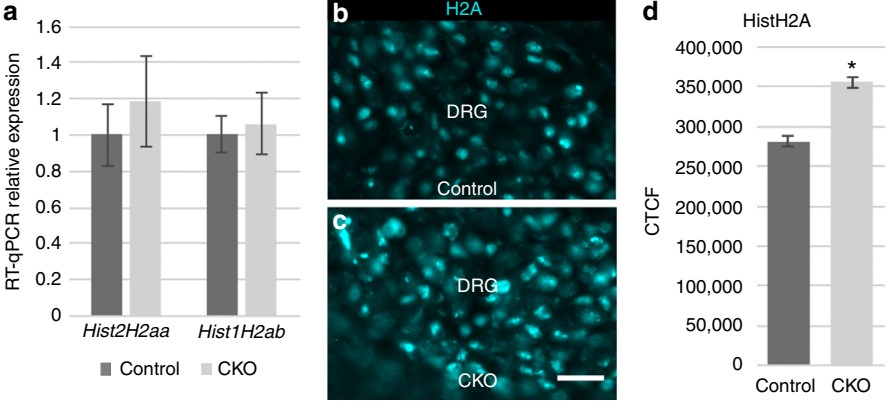

**Fig. 5** Multiple H2A histones are upregulated in Elongator CKO embryos. **a** RT-qPCR confirms normal transcription of *Hist2H2aa* ($P = 0.26$) and *Hist1H2ab* ($P = 0.81$). **b–g** Quantitative immunohistochemistry using an anti-HistH2A primary antibody confirms increased levels of H2A type histones ($P < 0.0001$); error bars denote SD in a and SEM in **d**; scale bar: 20 μm

confirm upregulation at the protein level via quantitative IHC (Fig. 5). Although we did not detect significant misregulation of H2B histones, the extreme AG bias exhibited by the majority of both *H2a* and *H2b* genes suggests that histones may represent a family of functionally-related genes whose expression is coordinated by Elongator[20].

**Upregulation of UPR genes in *Ikbkap* CKO embryos**. Loss of Elongator function in both yeast and mice has also been shown to cause ribosomal pausing and translational stress, which in turn triggered the unfolded protein response (UPR)[27,38]. Consistent with these findings, our transcriptome and proteome studies

indicate similar changes induced by loss of *Ikbkap/Elp1* in the developing PNS. Transcript levels of UPR genes *Atf5*, *Ddit*, *Atf4*, *Egr1*, *Gadd45a*, *Herpud1*, *Trib3*, and *Eif2s2* were all significantly increased (GSE80130); out of these eight upregulated genes, three of them were also detected in our proteome analysis; DDIT3, EGR1, and EIF2S2 were increased 9.7, 10.8, and 2.1-fold, respectively (not shown).

**Gene Ontology analysis of misregulated proteins**. We determined Gene Ontology (GO) biological processes for genes that were strong candidates for regulation by Elongator including downregulated proteins encoded by transcripts with an AA:AG >

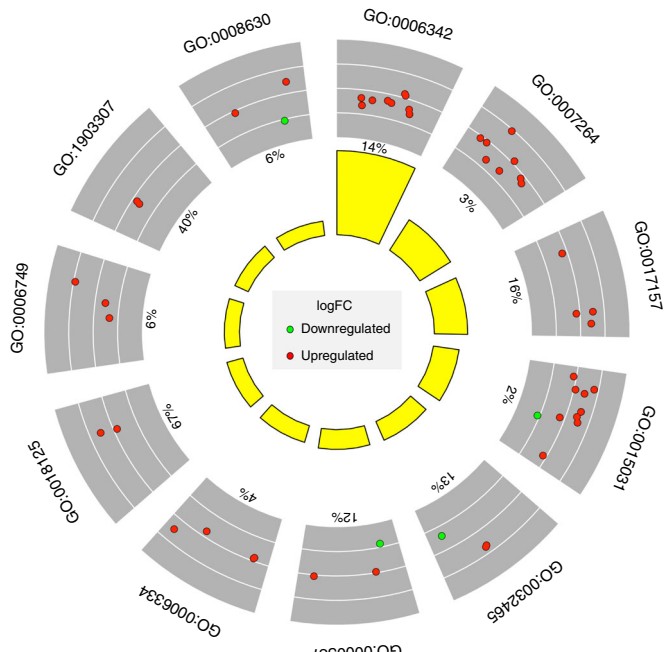

| GO ID | GO term | Genes | p-value |
|---|---|---|---|
| GO:0006342 | Chromatin silencing | *Hist1h2ab, Hist1h2aa, Hist1h2ad, Hist2h2ac, Hist1h2ah, Hist1h2ak, H2afx, Hist3h2a, Hist2h2aa1* | 3.26E−10 |
| GO:0007264 | Small GTPase-mediated signal transduction | *Rab3a, Rab3b, Rab3d, Rac3, Rab19, Rab4b, Rasl12, Rab20* | 8.35E−05 |
| GO:0017157 | Regulation of exocytosis | *Rab3a, Rab3b, Rab3d, Vamp2* | 1.79E−04 |
| GO:0015031 | Protein transport | *Rab3a, Rab3b, Rab3d, Tmed3, Ap2s1, Rab4b, Vamp3, Vamp2, Rab20, Gcc2* | 1.19E−03 |
| GO:0032465 | Regulation of cytokinesis | *Pdxp, Brca2, Calm3* | 5.01E−03 |
| GO:0000387 | Spliceosomal snRNP assembly | *Tex16, Snrpd2, Lsm4* | 5.86E−03 |
| GO:0006334 | Nucleosome assembly | *Hist1h2bm, Hist1h1e, Hist1h2bh, Hist2h2aa1* | 1.10E−02 |
| GO:0018125 | Peptidyl-cysteine methylation | *Rab3b, Rab3d* | 1.32E−02 |
| GO:0006749 | Glutathione metabolic process | *Gstm4, Ethe1, Gstt1* | 1.99E−02 |
| GO:1903307 | Positive regulation of regulated secretory pathway | *Rab3a, Rab3d* | 2.19E−02 |
| GO:0008630 | Intrinsic apoptotic signaling pathway in response to DNA damage | *Bax, Brca2, Sfn* | 2.38E−02 |

**Fig. 6** Significantly enriched GO biological processes for genes that are strong candidates for Elongator regulation. Green circles represent downregulated proteins encoded by genes with ≥ 1,005 codons and an AA:AG ratio > 1.3. Red circles represent upregulated proteins encoded by genes with ≤ 300 codons and an AA:AG ratio < 0.3. % Numbers indicate the percentage of proteins within a given GO category that are misregulated. Yellow bars represent the − log10 adjusted P-value. *All genes were transcribed normally

1.3 and size ≥ 1,005 codons, as well as upregulated proteins with an AA:AG bias of < 0.3 and transcript size ≤ 300 codons (Fig. 3, Supplementary Data 9). Interestingly, the GO group most misregulated in CKO neurons was chromatin silencing (GO:0006342), the same group characterized as being significantly AG-biased in budding yeast[20]. Additional highly misregulated GO groups included apoptotic signaling in response to DNA damage and groups with such key neuronal functions as small GTPase mediated signal transduction, protein transport, regulation of exocytosis, and regulation of the secretory pathway (Fig. 6 and Supplementary Fig. 4).

## Discussion

In addition to FD, which results from a splice site mutation in *IKBKAP/ELP1*, allelic variation in other Elongator subunits is associated with ALS (*ELP3*), intellectual disability (*ELP 2 and 4*), autism spectrum disorder (*ELP4*), and Rolandic epilepsy (*ELP4*)[39–42]. This study provides the first insight into how Elongator and codon bias may contribute to both the neurodevelopmental and neurodegenerative aspects of these diseases. We describe a novel mechanism for impacting gene expression directly at the protein level and demonstrate that the absence of this mechanism leads to misregulation of cellular pathways that are essential to neuron development, function, and survival. In total, our findings indicate that codon bias serves as a significant mechanism for regulating protein output, with grave cellular repercussions when the ability to interpret codon bias is compromised.

Importantly, our finding that Elongator loss leads to reduced expression of *Brca2* and other genes needed to repair DNA damage sheds new light on the pathogenesis of FD. An important hallmark of the disease is a failure in development of the PNS; autopsy studies on infants show that the small diameter pain and temperature subset of DRG neurons (TrkA+ nociceptors and thermoceptors) are selectively depleted, whereas large diameter

DRG neurons (TrkC+ proprioceptors) are less affected[25]. An outstanding question in the FD field has been to understand why the small diameter TrkA+ population exhibits this increased susceptibility. A major difference between the two subsets is that TrkC+ neurons are born early, during a limited first wave of neurogenesis, whereas TrkA+ cells are born during a second, extended wave of neurogenesis from mitotically active progenitors that go through numerous rounds of cell division to generate more than 70% of the final DRG neuron population[34,43–45]. We previously showed that these specific neural progenitors (Pax3+), as well as fully differentiated TrkA+ neurons, selectively exhibit elevated levels of apoptosis in the absence of *Ikbkap/Elp1*[22]. The data presented here suggest that the decreased number of small diameter TrkA+ neurons in FD likely results from DNA damage that accumulates in mitotically active progenitor cells as they attempt to generate the massive nociceptor and thermoceptor populations. This conclusion is supported by studies showing a specific requirement for *Brca2* and other homologous recombination repair genes during neural progenitor proliferation in the central nervous system[33,46]. As *Vcan* encodes an extracellular matrix proteoglycan that also functions in neuronal proliferation and differentiation, its depletion may exacerbate defects in neurogenesis imposed by compromised DNA damage response pathways[47,48].

An accumulation of damaged DNA over time could also contribute to the neurodegenerative hallmarks of FD including the progressive gait ataxia and death of retinal ganglion cells as patients age[49,50], as well as to the progressive neuronal apoptosis observed in FD model mice[51]. Genome maintenance poses a unique challenge for terminally differentiated neurons that go through their final cell division during embryogenesis and must therefore maintain genome integrity for many decades in spite of the thousands of DNA damage events they experience. As HDR is not accessible to post-mitotic cells, neurons depend on NHEJ and the more error-prone MMEJ pathway. If compromised expression of codon biased genes such as *Rif1* were to shift neuronal

DNA repair toward a heavier use of MMEJ, accumulated genetic damage would ensue. Indeed, an accumulation of DNA damage as a consequence of unchecked MMEJ is suspected to be the basis for neurodegeneration and tumor formation in ataxia telangiectasia[52].

A compromised ability to deal with DNA damage could also clarify why individuals born with FD are more than twice as likely to develop cancer during their shortened life span (40 years) compared with the general population, as well as why cancer onset occurs at a young age; the average age of tumor diagnosis in FD is < 30 years, whereas cancers in non-FD children and young adults are very rare (~ 3.5%)[53]. The highly codon biased nature of multiple DNA damage response genes likely also explains why 20% of cancers occurring in FD patients are of neural crest origin, the cellular lineage in FD with the lowest levels of correctly spliced *IKBKAP* transcript[24,53]. Independent of FD, the Elongator subunit *ELP3* has also been characterized as a tumor suppressor gene in studies correlating diminished ELP3 levels with a poor prognosis in cancer[54–56]. Thus, our findings provide a specific molecular pathway via which ELP3, the catalytic subunit of Elongator, may accomplish this tumor suppressor function.

Previous studies have demonstrated that the use of codon bias by families of functionally related genes helps coordinate and/or synchronize their expression[19,20]. It is also possible that by altering the levels of different tRNA modifications, a cell could quickly ramp up or slow down the production of an entire class of functionally related proteins, such as those needed to respond to DNA damage.

Encoded by mRNAs consisting of more than 3,300 total codons, both *Brca2* and *Vcan* fall within the largest 1% of all genes detected in our proteome analysis out of more than 17,000 genes (not shown). Their transcripts also contain an extreme number of AA-ending codons, 425 and 374, respectively, and are highly AA-biased (AA:AG ratio = 1.5 for both). Depletion of proteins with this specific genetic profile in the absence of Elongator has not been previously described, and demonstrates a connection between codon usage, transcript length, and tRNA modification in determining the expression levels of specific neuronal proteins. Post-transcriptional changes in less biased genes, as well as non-codon-biased genes were also present in our data set, suggesting that other, unidentified factors may function in concert with codon bias and transcript length to determine the overall impact of Elongator loss. Downstream effects from mis-regulation of other post-transcriptional pathways, such as the misregulation of numerous codon biased genes that function in ubiquitination (Supplementary Data 2), may also amplify the impact of elongator loss and at least in part explain the mis-regulation of non-codon-biased genes.

Recent studies have shown that acquisition of proper protein tertiary structure requires not only the correct amino acid sequence, but also a designated elongation rate as a growing peptide chain emerges from the ribosome[1–3]. Thus, for large AA-biased genes such as *Brca2* and *Vcan*, loss of Elongator would presumably lead to depleted protein levels due to both a slower translation rate of AA-ending codons, as well as polyubiquityla-tion and proteosomal degradation of misfolded proteins. Our data also support a model where altered translation rates and misfolding lead to ER stress and the UPR, which in turn could contribute to a loss of neuronal progenitors during development, as well as to the degeneration of mature neurons, both of which occur in mouse models of FD[3,22].

In total, our study is the first to demonstrate the ramifications of codon bias in neurons. Our work indicates that a delicate interplay between tRNA modification and the genetic ratio of AA-ending to AG-ending codons may fine tune the translational efficiency of specific neuronal transcripts. Although multiple codons may indeed encode the same amino acid, our findings demonstrate that synonymous codons are not equivalent, with codon selection playing a key role in translational efficiency. In summary, our work highlights a new stratum of the genetic code wherein the differential use of synonymous codons serves as a regulator of specific protein levels, and underscores how break-down of the machinery that reads this code can contribute to neurologic disease.

## Methods

**tRNA modification analysis**. DRG were collected and pooled from four control and four CKO embryos (E17.5) per biological replicate. Total RNA was extracted using a Qiagen miRNeasy Mini Kit. RNA was submitted to Arraystar (arraystar.com) for LC–MS analysis where it was processed as follows: total RNA samples were qualified by agarose gel electrophoresis and quantified using Nanodrop. tRNA was isolated from total RNA by Urea-PAGE, followed by hydrolysis to single nucleosides and dephosphorylation. The nucleoside solutions were deproteinized using Satorius 10,000 Da MWCO (molecular weight cut-off) spin filters. LC–MS analysis was performed on an Agilent 6460 QQQ mass spectrometer with an Agilent 1290 HPLC system using Multi reaction monitoring detection mode. Peak information of modified nucleosides for each sample was extracted using Agilent Qualitative Analysis software. Peaks with a signal-to-noise ratio ≥ 10 were considered as detectable nucleosides. Peak areas were then normalized to the quantity of purified tRNA for each sample.

**Production of codon-biased eGFP biosensors**. The mammalian optimized eGFP coding sequence, in which 42 out of 44 codons for Lys, Glu, and Gln are AG ending (AA:AG = 0.05) was used as the AG-biased biosensor (Vigene Biosciences, CV10002). For the AA-biased sensor, an eGFP construct in which all Lys, Glu, and Gln codons end in AA was commercially synthesized by Vigene Biosciences (vigenebio.com) as follows: ATGGTGAGCAAAGGCGAAGAACTGTTCACCGG GGTGGTGCCCATCCTGGTCGAACTGGACGGCGACGTAAACGGCCACAA ATTCAGCGTGTCCGGCGAAGGCGAA GGCGATGCCACCTACGGCAAACT GACCCTGAAATTCATCTGCACCACCGGCAAACTGCCCGTGCCCTGGCCC ACCCTCGTGACCACCCTGACCTACGGCGTGCAA TGCTTCAGCCGCTACC CCGACCACATGAAACAACACGACTTCTTCAAATCCGCCATGCCCGAAGG CTACGTCCAAGAACGCACCATCTTCTTCAAAGACGACGGC AACTACAAA ACCCGCGCCGAAGTGAAATTCGAAGGCGACACCCTGGTGAACCGCATCG AACTGAAAGGCATCGACTTCAAAGAAGACGGCCAACATCCTGGGGCACAA AACCGCACTGGAATACAACTACAACAGCCACAACGTCTATATCATGGCC GACAAACAAAAAAACGGCATCAAAGTGAACTTCAAAATCCGCCACAACA TCGAAGACGGC AGCGTGCAACTCGCCGACCACTACCAACAAAACACCC CCATCGGCGACGGCCCCGTGCTGCTGCCCGACAACCACTACCTGAGCA CCCAATCCGCCCTGAGCAAA GACCCCAACGAAAAACGCGATCACATGG TCCTGCTGGAATTCGTGACCGCCGCCGGGATCACTCTCGGCATGGACGA ACTGTACAAATAA. eGFP constructs were commercially packaged in a lentiviral vector with a PGK promoter driving their expression (vigenebio.com).

**In vitro neuronal cultures and lentiviral infection**. For culturing of DRG primary neurons, DRG from E17.5 embryos (*Wnt1-Cre; Ikbkap*[+/LoxP] X *Ikbkap* [LoxP/LoxP]) were collected into 1.5 ml microcentrifuge tubes containing neural basal media on ice. Following genotyping, DRG from five control (*Ikbkap*[+/LoxP]) and five CKO (*Wnt1-Cre; Ikbkap*[LoxP/LoxP]) embryos were pooled into 15 ml conical tubes containing neural basal media. Cells were spun down at 1000 r.p.m. for 1 min at 4 °C. After decanting the neural basal media, cells were incubated in 2 ml of 0.05% trypsin + EDTA (Thermo Fisher) at 37 °C for 15 min. Cells were then spun down as above and the trypsin quenched by addition of 2 ml 10% fetal bovine serum. After a final spin, cells were brought up in complete media containing 2% B-27, 2 nM L-Glutamine, 10 ng ml$^{-1}$ NGF, 10 ng ml$^{-1}$ NT3, and 10 ng ml$^{-1}$ BDNF (brain-derived neurotrophic factor) in neural basal media and triturated 20 times using pulled Pasteur pipettes of varying bore sizes. Once dissociated, cells were counted using a hemocytometer and plated at ~ 50,000 cells per well in an 8-well glass bottom chamber slide. Lentivirus expressing either AG-biased eGFP or AA-biased eGFP was added to both control and CKO cultures at a concentration of 500 viral particles per neuron.

**Imaging and corrected total cell fluorescence measurements**. Before conducting the codon-biased eGFP experiments, IHC of cultured neurons and glia using the primary antibodies anti-Tuj1 (neurons) and anti-S-100 (Schwann cells) was used to verify that cell morphology could be definitively used to distinguish between neurons and glia. For antibody details, see below. In these preliminary experiments, 98.9% of cells were identified correctly.

After lentiviral infection and a 48 h incubation at 37 °C (5% CO$^2$), a random number generator was used to randomly select 50 live eGFP-positive neurons from each culture. Neurons versus Schwann cells were identified based on morphology (see above). The 50 randomly selected neurons were imaged using a Leica TCS SP8 Confocal Laser Scanning microscope with identical laser intensity, gain, and offset

settings used for all cells. Cell size was measured using Image J Image Processing and Analysis Software (https://imagej.nih.gov/ij/). The average cell areas for control neurons expressing AG- and AA-biased eGFP were 893.0 (range = 359–1,598) and 784.9 (range = 345–1,429) square pixels, respectively. Average cell areas for CKO neurons expressing AG- and AA-biased eGFP were 575 (range = 368–858) and 598 (range = 381–854) square pixels, respectively. Image J was then used to measure the corrected total cell fluorescence (CTCF) of all cells with an area between 400–800 square pixels (~ 30 cells per genotype and construct). Data are presented as the average CTCF ± SEM.

**Bioinformatics**. To calculate the average occurrence of AA- and AG-ending codons and AA:AG ratios, the mouse and human coding sequences (versions Mm104 and Hs105, respectively) were downloaded from the Consensus CDS database (ftp.ncbi.nlm.nih.gov/pub/CCDS). Bioinformatics software Ana-conda (http://bioinformatics.ua.pt/software/anaconda/) was used for codon counts. Overall percentages and AA:AG ratios were calculated in Excel. The AA:AG ratio is calculated by dividing the number of AA-ending codons by the number of AG-ending codons.

**Proteomics**. DRG from seven control ($Ikbkap^{+/LoxP}$) and seven CKO ($Wnt1$-$Cre$; $Ikbkap^{LoxP/LoxP}$) embryos were collected on ice and snap frozen in liquid nitrogen. Global proteomic profiling was performed by Bioproximity (Bioproximity.com) as follows:

Samples were prepared for digestion using the filter-assisted sample preparation method (https://doi.org/10.1038/nmeth.1322). Briefly, the samples were suspended in 8 M urea, 50 mM Tris-HCl, pH 7.6, 3 mM dithiothreitol, sonicated briefly, and incubated in a Thermo-Mixer at 40 °C, 1,000 r.p.m. for 20 min. Samples were centrifuged to clarify and the supernatant was transferred to a 30 kDa MWCO device (Millipore) and centrifuged at 13,000 $g$ for 30 min. The remaining sample was buffer exchanged with 8 M urea, 100 mM Tris-HCl, pH 7.6, then alkylated with 15 mM iodoacetamide. The urea concentration was reduced to 2 M. Samples were digested using trypsin at an enzyme to substrate ratio of 1:100, overnight, at 37 °C in a Thermo-Mixer at 1,000 r.p.m. Digested peptides were collected by centrifugation.

Peptide desalting: a portion of the digested peptides, about 20 μg, were desalted using C18 stop-and-go extraction (STAGE) tips (https://doi.org/10.1038/nprot.2007.261). Briefly, for each sample a C18 STAGE tip was activated with methanol, then conditioned with 60% acetonitrile, 0.5% acetic acid followed by 2% acetonitrile, 0.5% acetic acid. Samples were loaded onto the tips and desalted with 0.5% acetic acid. Peptides were eluted with 60% acetonitrile, 0.5% acetic acid, and lyophilized in a SpeedVac (Thermo Savant) to near dryness, ~ 2 h.

For LC–MS/MS, each digestion mixture was analyzed by UHPLC–MS/MS. LC analysis was performed on an Easy-nLC 1000 UHPLC system (Thermo). Mobile phase A was 97.5% MilliQ water, 2% acetonitrile, and 0.5% acetic acid. Mobile phase B was 99.5% acetonitrile and 0.5% acetic acid. The 240 min LC gradient ran from 0% B to 35% B over 210 min, then to 80% B for the remaining 30 min. Samples were loaded directly to the column. The column was 50 cm × 75 μm I.D. and packed with 2 μm C18 media (Thermo Easy Spray PepMap). The LC analysis was interfaced to a quadrupole-Orbitrap mass spectrometer (Q-Exactive, Thermo Fisher) via nano-electrospray ionization using a source with an integrated column heater (Thermo Easy Spray source). The column was heated to 50 °C. An electrospray voltage of 2.2 kV was applied. The mass spectrometer was programmed to acquire, by data-dependent acquisition, MS/MS from the top 20 ions in the full scan from 400 to 1200 m/z. Dynamic exclusion was set to 15 s, singly-charged ions were excluded, isolation width was set to 1.6 Da, full MS resolution to 70,000, and MS/MS resolution to 17,500. Normalized collision energy was set to 25, automatic gain control to 2e5, max fill MS to 20 ms, max fill MS/MS to 60 ms, and the underfill ratio to 0.1%.

Mass spectrometer RAW data files were converted to MGF format using msconvert (https://doi.org/10.1038/nbt.2377). Detailed search parameters are printed in the search output XML files. Briefly, all searches required 10 p.p.m. precursor mass tolerance, 0.02 Da fragment mass tolerance, strict tryptic cleavage, 0 or 1 missed cleavages, fixed modification of cysteine alkylation, variable modification of methionine oxidation, and expectation value scores of 0.01 or lower. MGF files were searched using the human sequence library. MGF files were searched using X!!Tandem (https://doi.org/10.1021/pr0701198) using both the native (https://doi.org/10.1093/bioinformatics/bth092) and k-score (https://doi.org/10.1093/bioinformatics/btl379) scoring algorithms, and by OMSSA (https://doi.org/10.1021/pr0499491). All searches were performed on Amazon Web Services-based cluster compute instances using the Proteome Cluster interface. XML output files were parsed and non-redundant protein sets determined using Proteome Cluster (https://doi.org/10.1002/pmic.200900370). MS1-based features were detected and peptide peak areas were calculated using OpenMS (https://doi.org/10.1186/1471-2105-9-163). Proteins were required to have one or more unique peptides across the analyzed samples with $E$-value scores of 0.01 or less. Proteome data are available at ProteomeXchange (www.proteomexchange.org/) (reference PXD007869).

**Transcriptome analysis**. Total RNA was isolated from four control ($Ikbkap^{+/LoxP}$) and three CKO ($Wnt1$-$Cre$; $Ikbkap^{LoxP/LoxP}$) mice. Total cellular RNA was

extracted using an RNeasy plus mini kit (QIAGEN, Valencia, CA) and its quality was assessed with an Agilent BioAnalyzer using RNA 6000 Nano Chips (Agilent Technologies, Santa Clara, CA). Total RNA was used to generate biotin-labeled cRNA with the Illumina TotalPrep RNA Amplification Kit. In short, 0.5 μg of total RNA was first converted into single-stranded complementary DNA with reverse transcriptase using an oligo-dT primer containing the T7 RNA polymerase pro-moter site and then copied to produce double-stranded cDNA molecules. The double-stranded cDNA was cleaned and concentrated with the supplied columns and used in an overnight in-vitro transcription reaction where single-stranded RNA (cRNA) was generated incorporating biotin-16-UTP. A total of 0.75 μg of biotin-labeled cRNA from each of four control and three CKO biological replicates was hybridized at 58 °C for 16 h to Illumina's Sentrix mouse[8] Expression Bead-Chips (Illumina, San Diego, CA). Each BeadChip has ~ 24,000 well-annotated RefSeq transcripts with ~ 30-fold redundancy. The arrays were washed, blocked, and the labeled cRNA was detected by staining with streptavidin-Cy3. Hybridized arrays were scanned using an Illumina BeadStation 500 × Genetic Analysis Systems scanner and the image data extracted using Illumina's GenomeStudio software, version 1.9.0. For statistical analysis, the expression data were filtered to include only probes with a consistent signal on each chip and an Illumina detection $p$-value < 0.02.

Correlation analysis, sample clustering analysis, and principal component analysis on all probes were performed to identify/exclude any possible outliers. The resulting data set was next analyzed with DIANE 6.0, a spreadsheet-based microarray analysis program using value statistics for Z-score reliability below 0.05 and mean background-corrected signal intensity > 0. Transcriptome data are available at Gene Expression Omnibus (www.ncbi.nlm.nih.gov/geo/) (reference GSE80130).

**Gene Ontology analyses**. The plots for Fig. 6 and Supplementary Fig. 2 were created using the GOplot R package (https://CRAN.R-project.org/package=GOplot)[57]. Significance was calculated using the DAVID 6.8 Functional Annotation Tool with the GO biological process direct database (https://david.ncifcrf.gov/)[58,59].

**Real-time quantitative PCR**. Real-time quantitative PCR (qPCR) was performed using the Viia-7 qPCR System by Applied Biosystems and TaqMan Gene Expression Assays from Thermo Fisher Scientific: Mm01218747 ($Brca2$), Mm01283063 ($Vcan$), Mm01185982, Mm04179654 ($Hist2H2aa$), and Mm00779772 ($Hist1H2ab$). Relative expression levels or fold change are shown normalized against Gapdh (TaqMan GeneExpression Assay: Mm99999915). DRG were collected from three E17.5 control ($Ikbkap^{+/LoxP}$) and three CKO ($Wnt1$-$Cre$; $Ikbkap^{LoxP/LoxP}$) embryos. RNA was extracted using the Qiagen RNeasy Mini Kit and cDNA synthesized via the High Capacity RNA-to-cDNA Kit (Applied Bio-systems). Each reaction was performed in triplicate. Bars represent SD calculated from three Delta Delta Ct Expression values for each genotype.

**Comet assays**. DRG were collected from three control and three CKO embryos at E12.5 and dissociated as above. Approximately 5,000 cells were plated per each of three wells per embryo. Comet assays were performed according to the manu-facturer's instructions (Cell Biolabs, OxiSelect Comet Assay Kit, catalog number STA-350). For analysis, each well was imaged and ~ 200 randomly selected comets were analyzed per embryo using the ImageJ OpenComet plugin.

**Immunohistochemistry**. All washing, blocking, secondary antibody, and post-fixation steps were performed at room temperature. All other steps were performed at 4 °C unless stated otherwise. Embryos were fixed in 4% paraformaldehyde/phosphate-buffered saline (PBS) for 2 h, rinsed in PBS, cryoprotected through a series of sucrose solutions in PBS (15%, 30%), incubated for 2 h in a 1:1 mixture of 30% sucrose and optimal cutting temperature (OCT) compound (Tissue-Tek, Torrance, CA), followed by 2 h in OCT. Embryos were then frozen and cryosec-tioned at 16 m. For immunostaining, slides were bathed in TBS (tris-buffered saline) for 10 min, followed by NGS block (10% normal goat serum, 1% glycine, 0.4% Triton X-100 in 30 mM Tris, 150 mM NaCl) for 1 h, and overnight incuba-tion in primary antibody (in NGS block). Slides were then rinsed in NGS block, incubated in Alexa Fluor secondary antibody (1:2,000 in NGS block) for 1 h, rinsed in 3:1 TBS:NGS block, and mounted in Prolong Antifade Diamond (Invitrogen, La Jolla, CA). Control and experimental embryos were cryosectioned on the same day and sections were typically incubated in primary antibody on the same day that they were sectioned.

Microscopy images were captured using a Nikon TE200 inverted microscope and were captured with a QImaging QICAM 12 bit Mono Fast 1394 Cooled camera and QCapture software. Identical exposure times, gain, and offset settings were used to capture control and experimental images.

Primary antibodies included the following: BRCA2 (Biorbyt, orb10203, 1:25), VCAN (Abcam, ab177480, 1:100), S-100 (Agilent Technologies, Z0311, 1:400), Tuj1 (Biolegend, 801202, 1:1,000), and Histone H2A (Biorbyt, orb127582, 1:200).

Secondary antibodies used were Alexa Fluor goat anti-rabbit 488 and goat anti-mouse 568, (Invitrogen, 1:2,000).

**Data analysis and statistics**. For fetal cryosectioning, two alternating series of slides with approximately twenty 16 m sections per slide were generated from the mid-lumbar axial level (10 total slides). Different antibodies were used for each series of five slides, which typically contained sections from three to four DRG. For each series, the center-most section of each of 6 DRG (3 from the left half of the body and 3 from the right half of the body) was determined, and the 2 sections flanking that center section were imaged for a total of 12 sections per embryo. Identical gain and offset parameters were used for control and experimental slides and a minimum of three control and three CKO embryos were analyzed per antibody. CTCF was measured using Image J and the 10 brightest cells per section were averaged. Data are presented as the average CTCF ± SD. Statistical significance was determined by an unpaired Student's $t$-test.

**Mice**. The generation of *Wnt1-Cre; Ikbkap*$^{LoxP/LoxP}$ (*Ikbkap* CKO) embryos has been previously described[22]. *Wnt1-Cre* mice were purchased from the Jackson Laboratory (stock number 003829) and all strains were maintained on a C57BL/6 J background. For all analyses, *Wnt1-Cre; Ikbkap*$^{LoxP/LoxP}$ E17.5 embryos were used as experimental and *Ikbkap*$^{+/LoxP}$ littermates were used as controls. All genotyping was performed via routine PCR[22]. Ikbkap CKO and wild-type alleles were distinguished using the following primers: forward (F), 5′-GCACCTTCACTCCT-CAGCAT-3′ and reverse (R), 5′-AGTAGGGCCAGGAGAGAACC-3′. The presence of the Wnt1-Cre allele was detected using the following primers: F, 5′-GCCAATCTATCTGTGACGGC-3′ and R, 5′-CCTCTATCGAACAAGCATGCG-3′. All experiments were performed according to the National Institutes of Health Guide for Care and Use of Laboratory Animals and protocols were approved by the Montana State University Institutional Animal Care and Use Committee.

**Data availability**. The transcriptome data generated and analyzed in this study are available at Gene Expression Omnibus, accession number GSE80130 (https://www.ncbi.nlm.nih.gov/geo/query/acc.cgi?acc=GSE80130). Proteome data are available in the ProteomeXchange repository, accession number PXD007869 (http://massive.ucsd.edu/ProteoSAFe/QueryPXD?id=PXD007869).

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

## Acknowledgements

We thank the private benefactor whose donations to the MSU Billings Foundation, Familial Dysautonomia Research Fund made this work possible. L.G. was further funded by the National Institutes of Health (NIH), R15NS090384. Partial support was provided by the National Institute of General Medical Sciences of the NIH, P20GM103474 (L.G., J.F., R.B., and J.W.), the Intramural Research Program of the NIH, National Institute on Aging (K.B.), and NIH R01NS086796 (F.L.).

## Author contributions

L.G. conceptualized and designed the experiments, and supervised the project. L.G. and J.G. designed the methodology. F.L., Y.Z., E.L. and K.G.B. performed the transcriptomics. J.G. performed the bioinformatics. L.G., J.G., F.L., M.C., J.F., R.B. and J.W. collected and analyzed data. L.G. wrote the original draft. F.L. and J.G. provided review and editing. L.G. and J.G. prepared the figures.

## Additional information

**Competing interests:** The authors declare no competing financial interests.

