## [Peer Review File · Nature Communications]

Reviewers' comments:

Reviewer #1 (Remarks to the Author):

This is an important study, and i am generally enthusiastic.

It has been known for a very long time that codon usage can impact translation efficiency, and there is some considerable effort that has gone to understand how modifications of tRNAs can impact efficiency of AA vs AG ending codons. But in mammals, indeed in metazoans, there still is not much known about how this impacts biology. And in neurons there is still less known. This study takes as a launching point the finding that familial dysautonomia (FD), a rare genetic disorder, results from a defect in the elongator complex, which is required for both mcm5 modification and thiolation of the wobble U.

The work is pretty straight forward, but elegant. The authors test the impact of mutations elongator subunits as a model for FD. First a simple reporter system was used in which codon use of a GFP reporter was used neuronal culture of either control or KO neurons from floxed/CRE expressing embryos. The reporter was engineered to be either enriched for AG or for AA codons. Second, the authors used proteomics and transcriptomics to conduct a broad survey of protein and RNA expression levels as a function of codon bias and transcript length. Together, these experiments reveal that elongator disruption preferentially impacts the translation of long AA enriched transcripts. And conversely, it was found that short AG enriched transcripts have a tendency to exhibit increased expression.

The above phenomenology is generally consistent with the findings from the GFP reporter given that the reporter is short in length (thus it exhibits no change in the AA enriched version and shows an increased expression in the AG enriched version). In an ideal world, it would be nice for the authors to test effects on a longer reporter gene where the prediction is that the AA enriched version would show decreased expression. If an appropriately long reporter exists (i cant think of one immediately), this would be a nice addition. But on balance, I would classify this as "nice but not necessary".

Overall, i think this is a well designed study. The experiments are generally well controlled, and the findings are important and relevant.

The study does leave many questions unanswered. And while answering some of these is beyond the scope of this study, i think some discussion of a few points is warranted.

First, can the authors speculate why it would be advantageous (and conserved!) to organize the coding sequences of some classes of genes such as DNA damage response to have highly biased codon usage? I can imagine reasons, it may provide the cell with the means to switch to a program of 'emergency repair' by altering the modifications on tRNAs. But would this be a rapid or a very slow process given tRNA stability? Not sure. In any case, i think it would be a useful thing to speculate/comment on.

Second, the authors discuss the neurodegenerative phenotypes of FD a bit, and relate their findings back to neurodegenerative disorders such as ALS. But unlike the developmental

phenotypes of FD, neurodegenerative disorders are largely impacting post mitotic neurons. Such neurons do not have access to homologous repair of double stranded breaks. Instead, they can use NHEJ, or just head for apoptosis. I think this deserves some comment too if the authors want to include their discussion of implications for neurodegenerative disorders.

Finally, a minor issue: many of the figures had formatting issues with the labeling of both axes. The axes were often scrambled text. I was typically able to work out or guess what was being plotted, but obviously this needs to be fixed.

Reviewer #2 (Remarks to the Author):

NCOMMS-17-06808A-Z

Deciphering a secondary genetic code in neurons: The role of Elongator and codon bias in neurological disease

Remarks to authors:

This manuscript describes how mutations in IKBKAP/ELP1, a component of the Elongator complex, lead to FD (familial dysautonomia). The Elongator complex catalyzes the mcm5s2U34 modification at the wobble position of mammalian tRNA^{Lys}, tRNA^{Glu}, and tRNA^{Gln} to enable better reading of AA-ending codons while curtailing AG-ending codons. The central hypothesis is interesting in that the wobble modification is important for controlling translation and thus providing a second layer of regulation after the first layer of reading the genetic code. While the major conclusion is potentially interesting, the paper does not provide sufficient explanations to understand its data.

1. Figure 2: Even in control cells, where the *elp1* gene was intact, the AA-biased eGFP expression was lower relative to the AG-biased expression. It is not clear why. Also, in the AA-biased eGFP expression, there was no difference between the control and the conditional KO (CKO) cells. If the authors argued that this lack of difference is due to the small size of eGFP, it needs to be stated here. The last sentence of the paragraph describing Figure 2: "these data indicate that mammalian Elongator may also function in maintaining proper levels of protein encoded by AG-biased genes". Is it AG- or AA-biased genes?

2. Figure 3: All panels in this figure are poorly presented. All of the captions in these panels are too small to read and the presentation is not intuitive. The Figure legends do not help much either.

3. Figure 3b: The Y axis is "percent of proteins that are decreased". The authors stated that "55.0% of included proteins encoded by transcripts greater than 1755 codons with an AA:AG ratio greater than 1.5 are depleted in the CKO. While the end point of the line representing transcripts of more than 1755 codons with an AA:AG ratio of more than 1.5 reached 55%, this does not mean that these proteins are depleted. The same problem is with the 26.5% of protein with a transcript length greater than 505 codons and a ratio of

AA:AG of more than 1.5. Basically, authors need to explain why and how the Y axis represents protein depletion.

4. Figure 3c: the conclusion is that for proteins that are not translated with a dependence on the Elongator-mediated wobble modification, their levels go down as the AA:AG ratio goes up. While the data indicate that this is the case, but why? An explanation here will help readers understand the meaning of the results.

5. Figure 3d: the conclusion is that for proteins that are up-regulated upon depletion of the Elongator-mediated wobble modification, their levels stay the same as the AA:AG ratio goes up. Why is this?

6. Figure 3e: the conclusion is that when the Elongator-mediated wobble modification is lacking, the % of protein depleted remains the same even with an increase of AA:AG ratio. What does this mean?

7. The data on the translation of proteins that have AG-biased genes are inconsistent with the yeast *elp3* mutants. The authors' data suggest that the absence of the Elongator-mediated wobble modification would increase translation of such proteins, but the yeast *elp3* mutants (thus lacking the wobble modification) showed that the ribosome binding of tRNA lacking the modification was decreased. This inconsistency needs to be resolved.

Reviewer #3 (Remarks to the Author):

Goffena and colleagues present data showing changes in post-transcriptional protein levels based on the presence of the Elongator complex and on the AA/AG codon bias of transcripts. They performed quantitative mass spectrometry and RNA-seq on the dorsal root ganglia of mice with and without peripheral nervous system ablation of *Ikbkap/Elp1* expression, then, controlling for mRNA transcript levels, identified changes in protein levels associated with AA/AG codon bias. They also used a codon-biased GFP reporter construct to show differential GFP levels in *Ikbkap* null neuronal cultures.

Although the overall trends of the data support the model they present, they do not pay enough attention to addressing noise in their data (i.e., what the threshold is for a significant change in protein levels) or explaining data that do not fit their model. Additionally, the authors do not demonstrate a link between the presence of the Elongator complex, tRNA modifications (*mcm5* and *s2*), and the mouse phenotype. Moreover, although the authors' model for *IKBKAP*-mediated FD can explain some key phenomena of FD, they do not present enough data to support the connection between Elongator-regulated protein levels and the mechanism of FD.

Major and minor suggestions to improve the manuscript are outlined below:

1. It would be useful to have a brief discussion the phenotype of the conditional knock out

mice, and the aspects of familial dysautonomia it recapitulates. Please also briefly discuss how a conditional null allele compares to the FD splice site mutation that causes tissue-variable levels of exon skipping and premature protein termination.

2. It would be important to show that in the DRG knockout cells, the mcm5 and s2 modifications are missing. Additionally, does loss of these modifications in the CKOs lead to widespread protein aggregation, as has been shown in yeast and worm models?

3. Was the RNA for transcriptome analysis isolated from the dorsal root ganglia of embryos specifically, as was detailed for the proteomics?

4. Figure 3 needs to be simplified and presented in a way that the important take-home points are clear. Collapsing several of the graphs into one may be helpful; for example, for each AA:AG ratio, how many proteins increased or decreased? Please also address the quantity of the change in protein levels—what was considered a statistically significant (and biologically significant) change in protein levels? Additionally, Simplifying sentences when discussing Figure 3 will also help emphasize the important points. For example, “The percentage of proteins that are reduced at a post-transcriptional level in CKO embryos gradually increases with increasing AA bias” could become “In CKO embryos, increased AA bias in a gene is correlated with decreased protein levels.”

5. How do you explain the data that does not fit the model? For example, in Fig 3d, 20-25% percent of AA biased proteins increase expression levels in the CKOs. Additionally, in the filter to find long, heavily AA biased genes, fewer than half of the top 18 genes showed a decrease in expression in the CKOs (7 expressed normally and 4 over-expressed). If these genes are the most likely to be decreased when Elp1 is gone, why are so many of them unaffected or increased?

6. How well conserved are the wobble positions in proteins affected by Elp1 loss? Is use of the other codon tolerated in other species? Are there any SNPs with high frequency across human populations that would lead to use of the other codon?

7. Are the genes that fall within the parameters of Elongator dependence expressed in the same tissues that Elp genes are expressed in?

8. For Figure 4, the legend should read “Depleted levels of BRCA2 are correlated with elevated levels of DNA damage...” Additionally, this experiment would benefit from a positive control. How does the amount of DNA damage (comet tail length) in the CKOs compare to that of BRCA2 mutants (either null alleles or hypomorphic alleles)? It would be important to show that the DNA damage is BRCA2 dependent (i.e., will adding extra BRCA2 ameliorate the damage?).

9. Similarly, because this paper posits that an Elongator-based misregulation of proteins important for TrkA+ cells causes FD, a more thorough exploration of the effects of Elongator on these proteins would greatly strengthen the disease relevance of this paper. Validating that Elongator decreases the amount of other key proteins in these pathways (not just BRCA2) is needed, and showing that a phenotype can be improved by overexpressing these genes in the presence of Elongator mutations would be convincing.

10. In the Discussion, please note what is known about why mutations in different components of the same complex lead to different diseases (especially since some seem CNS, not PNS, specific). Is there similar evidence to suggest an effect on post-transcriptional regulation for these members of the Elongator complex? Do these proteins play other roles in addition to this complex?

11. Additionally, a number of the claims made in the discussion are not supported by the

methods/data in the rest of the paper (i.e. "We demonstrate...fine tunes the translation rate of specific neuronal transcripts," but although changes in protein levels are presented, no data is presented for translation rate.)

Reviewer #4 (Remarks to the Author):

This is a very interesting manuscript which attempt to decipher the roles of ELP1 subunit of the Elongator complex in regulating the translation of codon-biased genes in dorsal root ganglions of mice. The authors have appropriately made use of large scale OMICS technology to address this question and this idea will be beneficial to the community at large. Although this manuscript carries the title: "Deciphering a secondary genetic code in neurons: the role of Elongator and codon bias in neurological disease", this study mainly focuses on OMICS experiments and is hence heavily descriptive. There is not much detailed experiments being performed to unravel the precise link and the exact mechanisms of how this second genetic code could have precisely cause familial dysautonomia (FD). In that, may be the authors should modify the title so as to tone down a little, or to perform more validation experiments so as to tease apart unambiguously, the precise link among codon-biasness, the roles of ELP1 gene and FD. The following are my comments:

(i.) Page 4: Figure 2: The p-value for AA-biased control versus AA-biased CKO = 0.89. Is "0.89" a typo error? Or does this mean the conclusion drawn for such observation is not valid? What is the sample size for this eGFP biosensor study?

(ii.) Page 5, for Table S1, Among the 6474 genes reported, the fold change values for some of the transcripts are not < 1.5 folds and > -1.5 folds, as reported by the authors.

(iii.) From Table S1, it seems the authors have only 2 replicates for transcript analysis and the FC values for some transcripts are not consistent between the 2 replicates. For example, for Gad1 gene, the FC value is -2.86 for replicate 1 and 1.16 for replicate 2. Also, I presume the FC values reported have been log-transformed. However, the authors did not indicate so in Table S1.

(iv.) Continuing from (iii.), can the authors clearly specify the Fold Change (FC) cut off used for both transcripts and proteins, as simply FC or the FC values after being log-transformed?

(v.) From Figure 3, it appears that the sizes of proteins, besides their respective AA:AG ratios; also affect their abundance in the CKO samples. This is an interesting observation. However, the authors did not provide a viable explanation as to why the sizes of proteins play a role in this phenomenon.

(vi.) On Page 9, According to the authors, "For large, AA-biased genes, the proteome dataset of normally transcribed genes was filtered for targets that consisted of > 1755 total

codons with an AA:AG ratio > 1.3 (Fig 3b). Of the 18 genes that remained, 7 were under-expressed (fold change < -2), 7 were expressed normally (fold change < 2.0 and > -2.0), and 4 were overexpressed (fold change > 2.0) (Table S5)."

- I have re-checked Table S5, what I found instead, is that: ...25 genes that remained, 8 were under-expressed (fold change < -2), 14 were expressed normally (fold change < 2.0 and > -2.0), and 3 were overexpressed (fold change > 2.0)

- Again, I re-checked Table S2, this time I obtained: ...19 genes that remained, 7 were under-expressed (fold change < -2), 8 were expressed normally (fold change < 2.0 and > -2.0), and 4 were overexpressed (fold change > 2.0)

Can the authors explain the discrepancy?

Methods:

(i.) "...centrifuged at 13 k g" should be "centrifuged at 13,000 g"

(ii.) "20 micrograms" >> "20g"

(iii.) For proteomics data, please deposit your RAW and processed data files into ProteomeXchange repository (<http://www.proteomexchange.org/>) and cite your reference number in the manuscript. This is a requirement for proteomics community. Please also do so for the transcript data.

Response to referees' comments for manuscript NCOMMS-17-06808A-Z:

We would like to sincerely thank the reviewers for their time, and for their thoughtful and constructive comments. We have addressed each of their concerns below, to the betterment of the manuscript.

Reviewers' comments:

Reviewer #1 (Remarks to the Author):

This is an important study, and I am generally enthusiastic. It has been known for a very long time that codon usage can impact translation efficiency, and there is some considerable effort that has gone to understand how modifications of tRNAs can impact efficiency of AA vs AG ending codons. But in mammals, indeed in metazoans, there still is not much known about how this impacts biology. And in neurons there is still less known. This study takes as a launching point the finding that familial dysautonomia (FD), a rare genetic disorder, results from a defect in the elongator complex, which is required for both mcm5 modification and thiolation of the wobble U.

The work is pretty straight forward, but elegant. The authors test the impact of mutations elongator subunits as a model for FD. First a simple reporter system was used in which codon use of a GFP reporter was used neuronal culture of either control or KO neurons from floxed/CRE expressing embryos. The reporter was engineered to be either enriched for AG or for AA codons. Second, the authors used proteomics and transcriptomics to conduct a broad survey of protein and RNA expression levels as a function of codon bias and transcript length. Together, these experiments reveal that elongator disruption preferentially impacts the translation of long AA enriched transcripts. And conversely, it was found that short AG enriched transcripts have a tendency to exhibit increased expression.

The above phenomenology is generally consistent with the findings from the GFP reporter given that the reporter is short in length (thus it exhibits no change in the AA enriched version and shows an increased expression in the AG enriched version). In an ideal world, it would be nice for the authors to test effects on a longer reporter gene where the prediction is that the AA enriched version would show decreased expression. If an appropriately long reporter exists (I can't think of one immediately), this would be a nice addition. But on balance, I would classify this as "nice but not necessary".

Overall, I think this is a well-designed study. The experiments are generally well controlled, and the findings are important and relevant.

The study does leave many questions unanswered. And while answering some of these is beyond the scope of this study, I think some discussion of a few points is warranted.

First, can the authors speculate why it would be advantageous (and conserved!) to

organize the coding sequences of some classes of genes such as DNA damage response to have highly biased codon usage? I can imagine reasons, it may provide the cell with the means to switch to a program of 'emergency repair' by altering the modifications on tRNAs. But would this be a rapid or a very slow process given tRNA stability? Not sure. In any case, I think it would be a useful thing to speculate/comment on.

- *We thank the reviewer for this thoughtful question and input. Previous studies have shown that the use of codon bias by functionally related families of genes helps to coordinate their expression. We have added these comments to the discussion as well as a sentence regarding the possibility that altering the level of tRNA modification could be used to ramp up or slow down the production of entire classes of functionally related proteins.*

Second, the authors discuss the neurodegenerative phenotypes of FD a bit, and relate their findings back to neurodegenerative disorders such as ALS. But unlike the developmental phenotypes of FD, neurodegenerative disorders are largely impacting post mitotic neurons. Such neurons do not have access to homologous repair of double stranded breaks. Instead, they can use NHEJ, or just head for apoptosis. I think this deserves some comment too if the authors want to include their discussion of implications for neurodegenerative disorders.

- *This is an excellent point. We now include a codon usage analysis of genes that specifically function in the NHEJ pathway and a discussion of how compromised expression of the identified genes could contribute to the neurodegenerative aspects of both FD and other neurodegenerative diseases linked to compromised Elongator function.*

Finally, a minor issue: many of the figures had formatting issues with the labeling of both axes. The axes were often scrambled text. I was typically able to work out or guess what was being plotted, but obviously this needs to be fixed.

- *We apologize for scrambled text in our figures. This was likely due to our use of a PDF format for our figures which may not have been compatible with different versions of Adobe. We have replaced all of our figures with jpeg files which should fix this problem.*

Reviewer #2 (Remarks to the Author):

NCOMMS-17-06808A-Z

Deciphering a secondary genetic code in neurons: The role of Elongator and codon bias

in neurological disease

Remarks to authors:

This manuscript describes how mutations in IKBKAP/ELP1, a component of the Elongator complex, lead to FD (familial dysautonomia). The Elongator complex catalyzes the mcm5s2U34 modification at the wobble position of mammalian tRNALys, tRNAGlu, and tRNAGln to enable better reading of AA-ending codons while curtailing AG-ending codons. The central hypothesis is interesting in that the wobble modification is important for controlling translation and thus providing a second layer of regulation after the first layer of reading the genetic code. While the major conclusion is potentially interesting, the paper does not provide sufficient explanations to understand its data.

1. Figure 2: Even in control cells, where the *elp1* gene was intact, the AA-biased eGFP expression was lower relative to the AG-biased expression. It is not clear why.

- *Yes, this shows that AA-ending codons are inherently harder to translate because of restrictive codon-anticodon interactions. We now include a sentence explaining this in the corresponding results section.*

Also, in the AA-biased eGFP expression, there was no difference between the control and the conditional KO (CKO) cells. If the authors argued that this lack of difference is due to the small size of eGFP, it needs to be stated here.

- *We now include a sentence in the results section explaining how the small size of the AA-biased eGFP likely prevents its expression from being compromised in CKO neurons.*

The last sentence of the paragraph describing Figure 2: “these data indicate that mammalian Elongator may also function in maintaining proper levels of protein encoded by AG-biased genes”. Is it AG- or AA-biased genes?

- *“AG-biased genes” is correct and what we had intended. Figure 2 shows that the expression level of AG-biased eGFP increases in the absence of Elongator. To make this clearer, we have added verbiage explaining how the mcm⁵s² modification present in the control may actually reduce the ability of U₃₄ to pair with G-ending (wobble) codons such that when the modification is missing, the efficiency of translating AG-ending codons increases.*

2. Figure 3: All panels in this figure are poorly presented. All of the captions in these panels are too small to read and the presentation is not intuitive. The Figure legends do not help much either.

- *We have reworked Figure 3. It now includes fewer data, all of the captions and labels are larger, and the presentation is more clearly explained in the legend.*

3. Figure 3b: The Y axis is “percent of proteins that are decreased”. The authors stated that “55.0% of included proteins encoded by transcripts greater than 1755 codons with an AA:AG ratio greater than 1.5 are depleted in the CKO. While the end point of the line representing transcripts of more than 1755 codons with an AA:AG ratio of more than 1.5 reached 55%, this does not mean that these proteins are depleted. The same problem is with the 26.5% of protein with a transcript length greater than 505 codons and a ratio of AA:AG of more than 1.5. Basically, authors need to explain why and how the Y axis represents protein depletion.

- *In Figure 3b, the Y axis shows the percentage of proteins that were depleted by a fold change of 2 or more in our proteome study, as a function of increasing transcript length and increasing AA-bias. For example, 55% of proteins > 1755 codons with an AA:AG ratio > 1.5 were depleted by a fold change of 2 or more. To make this clearer, we have explained in both the text and the legend, that decreased proteins are defined as being expressed at a fold change that was 2 or more lower than the control in our proteome analysis.*

4. Figure 3c: the conclusion is that for proteins that are not translated with a dependence on the Elongator-mediated wobble modification, their levels go down as the AA:AG ratio goes up. While the data indicate that this is the case, but why? An explanation here will help readers understand the meaning of the results.

- *Figure 3c shows that the maintenance of normal expression becomes increasingly dependent on Elongator as AA-bias and transcript length increases. In other words, fewer and fewer genes are expressed at normal levels as transcript length increases (presumably because large genes contain the highest numbers of AA-ending codons) and/or as AA-bias increases (which increases the number of AA-ending codons and decreases the number of AG-ending codons). We have added similar language to the Figure 3 text to make this clearer. In addition, the impact of AA-ending versus AG-ending codons on translational efficiency in the absence of Elongator is more clearly explained throughout the manuscript.*

5. Figure 3d: the conclusion is that for proteins that are up-regulated upon depletion of the Elongator-mediated wobble modification, their levels stay the same as the AA:AG ratio goes up. Why is this?

- *These data suggest that the relatively small percentage of proteins that have a higher than average AA:AG ratio (are AA-biased) and are upregulated in the absence of Elongator are upregulated due to a mechanism other than their codon usage. i.e. we do not see a relationship between an increasing AA:AG ratio and the percentage of proteins that are upregulated. Since we do not know what this mechanism is, and because these data are not essential to*

understanding the connection between codon bias and Elongator, we have removed Figure 3d.

6. Figure 3e: the conclusion is that when the Elongator-mediated wobble modification is lacking, the % of protein depleted remains the same even with an increase of AA:AG ratio. What does this mean?

- We believe the reviewer may have misunderstood this figure. The graph in Figure 3d (formerly 3e) shows the percentage of proteins that are depleted as a function of increasing AA number, but in the **absence** of bias. In other words, the graph shows the behavior of proteins as the number of AA-ending codons increases, but in the context of an equally large number of AG-ending codons. This shows that AA-bias (a high number of AA-ending codons in the context of a low number of AG-ending codons) is what determines Elongator dependence rather than AA number alone. We have explained this more clearly in the Figure 3 text and hope that the below sentence in the discussion effectively explains the meaning of these data.*

“In addition, these data also explain why AA abundance in the context of an equally high number of AG-ending codons, did not impact protein levels in our study (Fig. 3d); the decreased translation rate of AA-ending codons in the absence of Elongator may have been offset by an increased translation rate of an equally large number of AG-ending codons.”

7. The data on the translation of proteins that have AG-biased genes are inconsistent with the yeast *elp3* mutants. The authors' data suggest that the absence of the Elongator-mediated wobble modification would increase translation of such proteins, but the yeast *elp3* mutants (thus lacking the wobble modification) showed that the ribosome binding of tRNA lacking the modification was decreased. This inconsistency needs to be resolved.

- We have to kindly disagree with the reviewer on this point, assuming he/she is referring to Rezgui et al., 2013 (tRNA tK^{UUU} , tQ^{UUG} , and tE^{UUC} wobble position modifications fine-tune protein translation by promoting ribosome A-site binding, PNAS). In this paper, the authors examine ribosomal binding to tRNAs in *urm1* and *elp3* yeast mutants. Specifically, they examine *in vitro* ribosomal binding of tRNA UUU (unmodified U_{34}) to a synthetic AAA mRNA within the A site and show a 60% decrease in ribosomal binding (presumably due to poor codon-anticodon interactions) which aligns with our findings. The authors did not examine the binding of tRNA UUU to an AAG wobble codon (they only included a synthetic mRNA with an AAA codon). We are suggesting that increased translational efficiency results when an unmodified tRNA U_{34} binds to a **G-ending (wobble)** codon. Our data indicate that unmodified tRNAs UUU, UUG, and UUC bind their respective G ending wobble codons (AAG, CAG, and GAG) within the ribosomal A site more efficiently than the modified forms.*

Reviewer #3 (Remarks to the Author):

Goffena and colleagues present data showing changes in post-transcriptional protein levels based on the presence of the Elongator complex and on the AA/AG codon bias of transcripts. They performed quantitative mass spectrometry and RNA-seq on the dorsal root ganglia of mice with and without peripheral nervous system ablation of *Ikbkap/Elp1* expression, then, controlling for mRNA transcript levels, identified changes in protein levels associated with AA/AG codon bias. They also used a codon-biased GFP reporter construct to show differential GFP levels in *Ikbkap* null neuronal cultures.

Although the overall trends of the data support the model they present, they do not pay enough attention to addressing noise in their data (i.e., what the threshold is for a significant change in protein levels) or explaining data that do not fit their model. Additionally, the authors do not demonstrate a link between the presence of the Elongator complex, tRNA modifications (*mcm5* and *s2*), and the mouse phenotype. Moreover, although the authors' model for IKBKAP-mediated FD can explain some key phenomena of FD, they do not present enough data to support the connection between Elongator-regulated protein levels and the mechanism of FD.

Major and minor suggestions to improve the manuscript are outlined below:

1. It would be useful to have a brief discussion the phenotype of the conditional knock out mice, and the aspects of familial dysautonomia it recapitulates. Please also briefly discuss how a conditional null allele compares to the FD splice site mutation that causes tissue-variable levels of exon skipping and premature protein termination.

- *We have added both to the introduction.*

2. It would be important to show that in the DRG knockout cells, the *mcm5* and *s2* modifications are missing. Additionally, does loss of these modifications in the CKOs lead to widespread protein aggregation, as has been shown in yeast and worm models?

- *We agree that determining whether or not the *mcm5* and *s2* modifications are missing in the DRG of *Wnt1-Cre; Ikbkap^{LoxP/LoxP}* embryos is an important validation of our conditional knockout. Our revised manuscript now includes an LC-MS based tRNA modification analysis showing decreased levels of the *mcm5s2U* modification (please see Supplementary Figure 1).*
- *Although we have not directly looked for protein aggregation, the included data showing the upregulation of genes involved in the unfolded protein response pathway suggest this may be the case.*

3. Was the RNA for transcriptome analysis isolated from the dorsal root ganglia of embryos specifically, as was detailed for the proteomics?

- *Yes, this has been clarified in the Figure 3 section of the results.*

4. Figure 3 needs to be simplified and presented in a way that the important take-home points are clear. Collapsing several of the graphs into one may be helpful; for example, for each AA:AG ratio, how many proteins increased or decreased? Please also address the quantity of the change in protein levels—what was considered a statistically significant (and biologically significant) change in protein levels? Additionally, Simplifying sentences when discussing Figure 3 will also help emphasize the important points. For example, “The percentage of proteins that are reduced at a post-transcriptional level in CKO embryos gradually increases with increasing AA bias” could become “In CKO embryos, increased AA bias in a gene is correlated with decreased protein levels.”

- *We have eliminated 3 of the panels in Figure 3 to help clarify the take-home points. The number of proteins increased or decreased for each data point is available in Table S2. We believe that adding this information to the graphs would actually make them more complicated than rather than simplify them. In Figure 3, a change in the expression level (either up or down) of 2-fold or greater was considered statistically and biologically significant. This is explained in the text, and we have clarified this in the Figure 3 legend.*

5. How do you explain the data that does not fit the model? For example, in Fig 3d, 20-25% percent of AA biased proteins increase expression levels in the CKOs.

- *We agree that this is an important point to consider, and have added the following text to the discussion, “Post-transcriptional changes in less biased genes as well as non-codon-biased genes were also present in our dataset, suggesting that other, unidentified factors may function in concert with codon bias and transcript length to determine the overall impact of Elongator loss. Downstream effects from misregulation of other post-transcriptional pathways, such as the misregulation of numerous codon biased genes that function in ubiquitination, may also amplify the impact of elongator loss and at least in part explain the misregulation of non-codon-biased genes.”*

Additionally, in the filter to find long, heavily AA biased genes, fewer than half of the top 18 genes showed a decrease in expression in the CKOs (7 expressed normally and 4 over-expressed). If these genes are the most likely to be decreased when Elp1 is gone, why are so many of them unaffected or increased?

- *This could again be due to the misregulation of other post-transcriptional pathways, or to posttranscriptional compensatory mechanisms. We feel that addressing this question in more detail is beyond the scope of this paper.*

6. How well conserved are the wobble positions in proteins affected by Elp1 loss? Is use of the other codon tolerated in other species? Are there any SNPs with high frequency across human populations that would lead to use of the other codon?

- *This is an interesting question and one we would like to pursue, but we would assert that it is beyond the scope of this paper.*

7. Are the genes that fall within the parameters of Elongator dependence expressed in the same tissues that Elp genes are expressed in?

- *Yes, we have shown robust expression of *Ikbkap* (*Elp1*) in the dorsal root ganglia (George et al., 2013), the same tissue used for this proteome study. The introduction now includes a sentence that describes *Ikbkap* expression in the DRG.*

8. For Figure 4, the legend should read “Depleted levels of BRCA2 are correlated with elevated levels of DNA damage...”

- *Yes, this is a much more accurate reflection of our data and has been changed. Thank you.*

Additionally, this experiment would benefit from a positive control. How does the amount of DNA damage (comet tail length) in the CKOs compare to that of BRCA2 mutants (either null alleles or hypomorphic alleles)? It would be important to show that the DNA damage is BRCA2 dependent (i.e., will adding extra BRCA2 ameliorate the damage?).

- *Since several other DBR genes are codon biased and misregulated, comparing *Brca2* mutants would not likely serve as an appropriate control, nor would overexpressing *Brca2* necessarily ameliorate the damage.*

9. Similarly, because this paper posits that an Elongator-based misregulation of proteins important for TrkA+ cells causes FD, a more thorough exploration of the effects of Elongator on these proteins would greatly strengthen the disease relevance of this paper. Validating that Elongator decreases the amount of other key proteins in these pathways (not just BRCA2) is needed, and showing that a phenotype can be improved by overexpressing these genes in the presence of Elongator mutations would be convincing.

- *We propose that one class of genes regulated by Elongator are those needed for HDR and that the TrkA subset of DRG neurons are particularly vulnerable to misregulation of these genes because of the extended proliferation required to generate the TrkA population. We did not intend to suggest that misregulation of this class of proteins alone causes FD, but rather contributes to one specific hallmark of the disease (selective loss of the TrkA+ DRG subset). However, we*

agree that validation of decreased levels of other DBR genes would strengthen the paper. We now include data validating decreased protein levels of Setx (Senataxin), mutations in which cause juvenile onset ALS.

10. In the Discussion, please note what is known about why mutations in different components of the same complex lead to different diseases (especially since some seem CNS, not PNS, specific). Is there similar evidence to suggest an effect on post-transcriptional regulation for these members of the Elongator complex? Do these proteins play other roles in addition to this complex?

- To our knowledge, it is not currently known why mutations in different components of Elongator lead to different diseases, although the question is extremely interesting. It may be of interest to the reviewer however to know that FD and ALS actually share many overlapping phenotypes, including deterioration of both PNS and CNS populations in both diseases. Degeneration of sensory neurons in ALS has now been documented in multiple reports (Isaacs et al, 2007; Rabin et al., 1999). In addition, mice in which Ikbkap/Elp1 is ablated in the CNS show progressive degeneration of spinal cord motor neurons (Chaverra et al, 2017), and a progressive motor phenotype occurs in FD (including the loss of motor neurons in some autopsy reports (Dyck et al., 1978), although motor neuron degeneration has not been extensively investigated. Furthermore, FD is unique in that it results from mutation in a splice acceptor site with some tissues being more capable of correctly splicing the mutated pre-mRNA than others. Hence the different aspects of diseases associated with Elongator impairment may relate to the amount of functional Elongator complex present in different neuronal subsets. Since complete loss of any of the Elongator subunits is lethal, mutations that are tolerated likely cause differences in the amount or pattern of subunit expression, rather than a complete loss of function.*
- With regard to the reviewer's additional questions, ELP 3 has been conclusively linked to post-transcriptional regulation of codon biased genes (Bauer et al., 2012). Whether or not ELP proteins function in capacities independent of Elongator is a fascinating question, especially given the multiple splice forms of Ikbkap. We are actually currently investigating this question, however, a discussion on this subject is beyond the scope of this paper.*

11. Additionally, a number of the claims made in the discussion are not supported by the methods/data in the rest of the paper (i.e. "We demonstrate...fine tunes the translation rate of specific neuronal transcripts," but although changes in protein levels are presented, no data is presented for translation rate.)

- We acknowledge that claims in the discussion may have been overstated and we have tempered the language accordingly.*

Reviewer #4 (Remarks to the Author):

This is a very interesting manuscript which attempt to decipher the roles of ELP1 subunit of the Elongator complex in regulating the translation of codon-biased genes in dorsal root ganglions of mice. The authors have appropriately made use of large scale OMICS technology to address this question and this idea will be beneficial to the community at large. Although this manuscript carries the title: “Deciphering a secondary genetic code in neurons: the role of Elongator and codon bias in neurological disease”, this study mainly focuses on OMICS experiments and is hence heavily descriptive. There is not much detailed experiments being performed to unravel the precise link and the exact mechanisms of how this second genetic code could have precisely cause familial dysautonomia (FD). In that, may be the authors should modify the title so as to tone down a little, or to perform more validation experiments so as to tease apart unambiguously, the precise link among codon-biasness, the roles of ELP1 gene and FD. The following are my comments:

- *We acknowledge that our title may have been too bold and have modified it to the following:*

Deciphering a secondary genetic code in neurons: the role of codon bias in regulating neuronal protein levels and implications for Elongator-mediated neurologic disease

(i.) Page 4: Figure 2: The p-value for AA-biased control versus AA-biased CKO = 0.89. Is “0.89” a typo error? Or does this mean the conclusion drawn for such observation is not valid? What is the sample size for this eGFP biosensor study?

- *No, this is not an error. Our conclusion from this experiment is that the eGFP gene is too small at only 240 codons to be impacted by AA bias. We have added a sentence to the results section to make this point clearer.*

(ii.) Page 5, for Table S1, Among the 6474 genes reported, the fold change values for some of the transcripts are not <1.5 folds and > -1.5 folds, as reported by the authors.

- *We apologize to the reviewer for this confusion. In our original submission, we included individual transcriptome data from two separate experiments (each using 3 different biological replicates) as well as the average fold change between the two experiments. Although some of the fold change values for the individual experiments fell outside of this range, none of the averages were actually > 1.5, or < -1.5. However, this was obviously confusing.*
- *Since the data from only one of these transcriptome experiments was uploaded to Gene Expression Omnibus (GEO) (accession GSE80130), we have re-plotted all of our charts and made new tables that exclusively use the data from this experiment that is available at GEO. This data will become public in April, 2018. We have provided the editor with a secure token that allows read-only access to*

the accession while it remains in private status. In addition, the transcriptome data that is included in the supplementary tables is now presented in a much clearer fashion and we sincerely apologize for this confusion.

(iii.) From Table S1, it seems the authors have only 2 replicates for transcript analysis and the FC values for some transcripts are not consistent between the 2 replicates. For example, for Gad1 gene, the FC value is -2.86 for replicate 1 and 1.16 for replicate 2. Also, I presume the FC values reported have been log-transformed. However, the authors did not indicate so in Table S1.

- *Again, we apologize for this confusion. As explained above, we initially included transcriptome results from two separate experiments (with each experiment using 3 biological replicates), as well as the average of the two experiments. To simplify our analysis, we have regenerated all of our plots and tables exclusively using the data that is available at GEO.*

(iv.) Continuing from (iii.), can the authors clearly specify the Fold Change (FC) cut off used for both transcripts and proteins, as simply FC or the FC values after being log-transformed?

- *Protein fold changes were log transformed and are now labeled as “Protein pool log₁₀ intensity”. Transcript fold changes were not log transformed, and thus are simply designated as FC.*

(v.) From Figure 3, it appears that the sizes of proteins, besides their respective AA:AG ratios; also affect their abundance in the CKO samples. This is an interesting observation. However, the authors did not provide a viable explanation as to why the sizes of proteins play a role in this phenomenon.

- *We now include the following sentence in the results section, “Importantly, we also found that large AA-biased genes are more likely to be impacted by Elongator loss than are small AA-biased genes, presumably because large genes contain the highest numbers of AA-ending codons.”*

(vi.) On Page 9, According to the authors, “For large, AA-biased genes, the proteome dataset of normally transcribed genes was filtered for targets that consisted of > 1755 total codons with an AA:AG ratio > 1.3 (Fig 3b). Of the 18 genes that remained, 7 were under-expressed (fold change < -2), 7 were expressed normally (fold change < 2.0 and > -2.0), and 4 were overexpressed (fold change > 2.0) (Table S5).”

- I have re-checked Table S5, what I found instead, is that: ...25 genes that remained, 8 were under-expressed (fold change < -2), 14 were expressed normally (fold change < 2.0 and > -2.0), and 3 were overexpressed (fold change > 2.0)

- Again, I re-checked Table S2, this time I obtained: ...19 genes that remained, 7 were under-expressed (fold change < -2), 8 were expressed normally (fold change < 2.0 and > -2.0), and 4 were overexpressed (fold change > 2.0)

Can the authors explain the discrepancy?

- *We thank the reviewer for catching this error. For Table S5, we mistakenly included a table using an AA:AG ratio greater than 1.2 rather than 1.3. Tables S5 and S1 now include 18 genes that have 1755 or more total codons and an AA:AG ratio > 1.3, with 7 under-expressed, 7 expressed normally, and 4 overexpressed.*

Methods:

(i.) "...centrifuged at 13 k g" should be "centrifuged at 13,000 g"

- *Corrected, thank you.*

(ii.) "20 micrograms" >> "20g"

- *Corrected, thank you.*

(iii.) For proteomics data, please deposit your RAW and processed data files into ProteomeXchange repository (<http://www.proteomexchange.org/>) and cite your reference number in the manuscript. This is a requirement for proteomics community. Please also do so for the transcript data.

- *We have now deposited our proteome data files into the ProteomeXchange repository as requested (reference # PXD007869). We have provided the editor with a username and password to allow viewing of the dataset while it remains in private status.*
- *The transcriptome data is also available at Gene Expression Omnibus as described above (reference # GSE80130). We now also note these deposits in the methods section.*

REVIEWERS' COMMENTS:

Reviewer #1 (Remarks to the Author):

This manuscript has been revised and improved both in terms of the text and the data content. I already was quite enthusiastic about the first version, and the authors have now addressed all of my concerns. Very nice manuscript.

Reviewer #2 (Remarks to the Author):

This revised manuscript by Dr. George and colleagues has sufficiently addressed most of my previous concerns. The only remaining issue is in response to my comment #7. The authors assumed that tRNA with U34, when unmodified, binds to G-ending codons with higher translational efficiency relative to A-ending codons. This is an assumption without experimental data and it is not intuitively accurate. The publication referred to by the authors "Rezgui et al., 2013 (tRNA tKUUU, tQUUG, and tEUUC wobble position modification fine-tune protein translation by promoting ribosome A-site binding, PNAS) has detailed kinetic measurements and thus their conclusion is on a solid ground. In the absence of any kinetic data, the authors of this manuscript should tune down their claim.

Reviewer #3 (Remarks to the Author):

The authors should be commended for dramatically improving the manuscript based on the comments of all four reviewers. If (and only if) additional requests for revisions are made, the authors should consider:

1. Including a well-argued explanation for why fewer than 1/2 of the top 18 genes (filtered for heavily AA biased genes) showed reductions in the CKOs.
2. An assessment of the conservation of wobble positions in transcripts/proteins affected by Elp1 loss.

Reviewer #4 (Remarks to the Author):

This study is interesting and original in nature although it contained quite a number of careless mistakes in the first version of manuscript. This second version of the manuscript is clearer after the authors made the following changes:

- (i.) Improved on the diagrams and graphs
- (ii.) Inserted more explanation in the text to clear my confusion
- (iii.) corrected some of the mistakes resulting from the supplementary data

All my doubts and questions were also addressed. Therefore I support the acceptance of this manuscript.

Response to referees' comments for manuscript NCOMMS-17-06808B

Below we have addressed each of the reviewer's final concerns.

Reviewers' comments:

Reviewer #2:

This revised manuscript by Dr. George and colleagues has sufficiently addressed most of my previous concerns. The only remaining issue is in response to my comment #7. The authors assumed that tRNA with U34, when unmodified, binds to G-ending codons with higher translational efficiency relative to A-ending codons. This is an assumption without experimental data and it is not intuitively accurate. The publication referred to by the authors "Rezgui et al., 2013 (tRNA tKUUU, tQUUG, and tEUUC wobble position modification fine-tune protein translation by promoting ribosome A-site binding, PNAS) has detailed kinetic measurements and thus their conclusion is on a solid ground. In the absence of any kinetic data, the authors of this manuscript should tune down their claim.

- *Thank you, we have tempered our language accordingly.*

Reviewer #3:

The authors should be commended for dramatically improving the manuscript based on the comments of all four reviewers. If (and only if) additional requests for revisions are made, the authors should consider:

1. Including a well-argued explanation for why fewer than 1/2 of the top 18 genes (filtered for heavily AA biased genes) showed reductions in the CKOs.

- *We now include a brief discussion regarding possible explanations for the observed normal levels of more than half of our top AA-biased candidate genes.*

2. An assessment of the conservation of wobble positions in transcripts/proteins affected by Elp1 loss.

- *This is an interesting question and one we would like to pursue, but we would assert that it is beyond the scope of this present paper.*